# Mangrove consortium resistant to the emerging contaminant DEHP: Composition, diversity, and ecological function of bacteria

Julia de Morais Farias[1,2]☉, Leandro Araujo Argolo[3‡], Raquel A. F. Neves[1,4]☉, Natascha Krepsky[1,2]☉*, José Augusto P. Bitencourt[3,5]☉

**1** Graduate Program in Neotropical Biodiversity, Federal University of the State of Rio de Janeiro (UNIRIO), Rio de Janeiro, Rio de Janeiro, Brazil, **2** Laboratory of Water Microbiology (LACQUA), Department of Environmental Science, Institute of Biosciences, Federal University of the State of Rio de Janeiro (UNIRIO); Rio de Janeiro, Rio de Janeiro, Brazil, **3** Instituto Tecnológico Vale, Desenvolvimento Sustentável (ITV), Belém, Pará, Brazil, **4** Research Group of Experimental and Applied Aquatic Ecology, Department of Ecology and Marine Resources, Institute of Biosciences, Federal University of the State of Rio de Janeiro (UNIRIO), Rio de Janeiro, Rio de Janeiro, Brazil, **5** Rede de Biodiversidade e Biotecnologia da Amazônia Legal (BIONORTE), Brazil

☉ These authors contributed equally to this work.
‡ LAA also contributed equally to this work.
* natascha@unirio.br

## Abstract

The continuous use of Di(2-ethylhexyl) phthalate (DEHP) in plastic products turns it into a ubiquitous contaminant in the environment. However, DEHP can cause harm to human beings, wildlife, and ecosystems due to its estrogenicity and toxicity. Thus, finding an efficient approach to removing this contaminant from the environment is crucial. The present study aimed to prospect and characterize a bacterial consortium (MP001) isolated from a neotropical mangrove for DEHP bioremediation. A laboratory experiment was performed with environmentally relevant DEHP concentrations (0.05, 0.09, 0.19, 0.38, 0.75, 1.50, 3.00, and 6.00 mg L$^{-1}$) to determine the consortium resistance to this contaminant and high-throughput sequencing was accomplished to assess the bacterial composition, diversity, and potential ecological function of consortium MP001. The consortium MP001 presented a significant biomass increase throughout short-term incubations with increasing concentrations of DEHP (GLMs, p< 0.001). MP001 was constituted by *Paraclostridium* sp. (78.99%) and *Bacillus* sp. (10.73%). After 48 h of consortia exposure to DEHP, the bacterial population changed to *Paraclostridium* (50.00%), *Staphylococcus* sp. (12.72%), *Staphylococcus epidermidis* (10.40%) and *Bacillus* sp. (17.63%). In the negative control, the bacteria community was composed of *Paraclostridium* sp. (54.02%), *Pseudomonas stutzeri* (19.44%), and *Staphylococcus* sp. (11.97%). The alpha diversity of the MP001 consortium was not significant (Kruskall-Wallis; p > 0.05), and no significant difference was found between the DEHP treatment and the negative control. Furthermore, the potential ecological function found in the consortium MP001 with higher potential for

**Data availability statement:** All relevant data are within the manuscript and its Supporting Information files.

**Funding:** Authors are grateful for the research grants attributed by the Foundation Carlos Chagas Filho Research Support of the State of Rio de Janeiro (FAPERJ) to Raquel A. F. Neves (E-26/201.283/2021; E-26/210.024/2024), to Natascha Krepsky (E-26/211.470/2021), the Graduate Program in Neotropical Biodiversity (E-26/211.043/2021) and the research grant attributed by the Brazilian National Council for Scientific and Technological Development (CNPq) to Raquel A. F. Neves (PQ2; 306212/2022-6). This study was also financed by the Brazilian National Council for Scientific and Technological Development (CNPq), through the scholarship (master's degree) to Julia de Morais Farias. The funders had no role in study design, data collection and analysis, decision to publish, or preparation of the manuscript.

**Competing interests:** The authors have declared that no competing interests exist.

application in bioremediation purposes was fermentation. The results found in this study highlight the potential of a bacterial consortium to be used in the bioremediation of DEHP-contaminated aquatic environments.

## Introduction

Phthalate esters (PAEs) are a group of synthetic organic compounds that have become one of the most common pollutants in the world [1]. The global annual production of PAEs has been estimated at 5.5 million tons [2]. This large number of PAEs is used to enhance the flexibility and durability of polyvinyl chloride (PVC) plastics and are found in several products, including automotive, electrical, and medical devices, personal care products (PCPs), cosmetics, and food packages [3–5]. Due to the low cost and versatility, the most widely used PAE in manufacturing is the Di(2-ethylhexyl) phthalate (DEHP) [3]. The use of DEHP has been increasing and the global market of DEHP, which reached US$ 9.1 billion in 2022, is expected to expand by 4.1% from 2023 to 2031, reaching US$ 13.1 billion [3,6]. DEHP is used in plastic products as a chemical additive; thus, it is not covalently bound into polymer matrices and can migrate to the surrounding environment [7–8]. DEHP has already been found at environmentally relevant concentrations in several environments, including air, soil, sediment, and water [e.g., 2, 9–11]. Notwithstanding, this compound accumulates in the sediments rather than in surface waters [12–13]. Indeed, DEHP was detected in nine coastal and marine sediments of coastal areas of Rio de Janeiro (Brazil) [14].

Pollutants accumulate more easily in mangrove ecosystems than other coastal environments because of their unique geochemistry. Mangroves are a transitional coastal ecosystem between terrestrial and marine environments frequently exposed to contamination from river water, tides, and surface runoff [15]. Besides, mangrove sediment is rich in organic matter (OM) that is known to be associated with lipophilic organic contaminants, such as DEHP [16]. Thus, mangrove sediments could be a reservoir of DEHP and a secondary source of pollution of this compound [13,15]. However, DEHP is a hazardous compound known as an endocrine disruptor [17]. Because of their estrogenicity, teratogenicity, mutagenicity, and carcinogenicity, DEHP was listed as a priority pollutant by the United States Environmental Protection Agency [18], the European Union [19], and the China National Environmental Monitoring Center [20]. DEHP can also induce disrupting endocrine effects, oxidative stress, metabolic disorders, and toxicity to wildlife [15,21–27]. Because of the harm that DEHP can cause to humans and the environment, it is crucial to find an efficient approach to eliminate this pollutant from the environment.

An efficient, eco-friendly, and economical technology to remove or reduce pollutants from the environment is bioremediation. Bioremediation is an approach that uses biological systems, mainly microorganisms, to degrade contaminants [28–29]. Among them, bacteria are the most promising in the degradation, which occurs when they use pollutant molecules as energy and carbon sources for growth

[29–31]. Isolated bacterial strains have been reported as able to degrade DEHP [e.g., 31–38]. Nevertheless, bacterial consortiums can be more effective in degradation than isolated strains [39–40]. The co-metabolism and interaction between the species improve degradation capability and resistance to environmental pollutants. Thus, pollutant degradation can be remarkably effective when carried out by bacterial communities of consortiums [39–41]. Mangrove microbial communities play a crucial role in the biogeochemical and nutrient cycles and can change in the presence of pollutants, holding great potential for biodegradation of contaminated sites [42–44]. However, despite the detection of DEHP in *Avicennia schaueriana*, a typical mangrove tree found in Brazilian ecosystems [45], there is no data of a neotropical consortium able to be used in the DEHP bioremediation.

Therefore, this study focused on isolating a bacterial consortium with biotechnological potential for DEHP bioremediation from an impacted mangrove area. The analysis aimed to assess the bacterial resistance to DEHP, the shifts in microbial diversity from samples exposed to DEHP, and the implications on their potential ecological functions. This research represents a critical step toward developing an eco-friendly biotechnological solution using mangrove sediment bacteria for DEHP degradation, contributing to mitigating its harmful effects on aquatic ecosystems and promoting healthier environments.

## Materials and methods

### Chemicals

The solution of DEHP was purchased from Sigma-Aldrich, USA (purity 98%; CAS number: 117-81-7; EC number: 204-211-0). A stock solution was prepared diluting the commercial solution of the compound in distilled water to reach the concentration of $40\,mg\,L^{-1}$. The stock solution was conditioned in a glass flask, previously decontaminated using methanol, and stored at room temperature in the dark until experiments.

### Sampling and consortium isolation

The consortium was previously isolated from the superficial sediment of the Magé mangrove, located in Rio de Janeiro state (22°43'14"S e 43°11'20" W). The sampling station in the mangrove was chosen because it is surrounded by Guanabara Bay, a bay with a strong anthropogenic impact (Fig 1) [46]. For that, 10 g of the sediment was transferred to 100 mL of the culture medium containing: beef extract ($3\,g\,L^{-1}$), beef peptone ($5\,g\,L^{-1}$), sodium chloride (NaCl, $30\,g\,L^{-1}$), and sodium phosphate dibasic ($Na_2HPO_4$, $1\,g\,L^{-1}$), and incubated at 37°C. After the isolation, the consortium was enriched with 1 mL petroleum for previous experiments [47] and denominated consortium MP001. Aliquots of MP001 were preserved using glycerol 30% in the same proportion of consortium (1:1), and they were kept frozen (-20°C) for posterior incubations.

### Bacterial inoculum preparation

For each experimental trial, a bacterial inoculum was prepared with MP001 consortia grown in TSB media (casein peptone ($17g\,L^{-1}$), soy peptone ($3g\,L^{-1}$), dextrose ($2.5g\,L^{-1}$), sodium chloride (NaCl, $5g\,L^{-1}$), and dipotassium phosphate ($K_2HPO_4$, $2.5g\,L^{-1}$)). The consortium was harvested by centrifugation (Fanem excelsa II) at 4,300 rpm for 15 min at room temperature and washed once with a 0,9% saline buffer to remove impurities. 20 mL of saline buffer was added to the pellet and homogenized in a vortex for 30 seconds. Before each assay, the cell density of the bacterial inoculum (1 mL) was estimated using the McFarland scale (Probac), and bacteria biomass was measured through absorbance analysis (600 nm) using a Trilogy Laboratory Fluorometer (Turner Designs).

### DEHP resistance assays

The consortium MP001 was exposed to eight DEHP concentrations (0.05, 0.09, 0.19, 0.38, 0.75, 1.50, 3.00, and $6.00\,mg\,L^{-1}$) to assess microorganisms' resistance to the contaminant. Exposure concentrations were prepared by serial dilutions

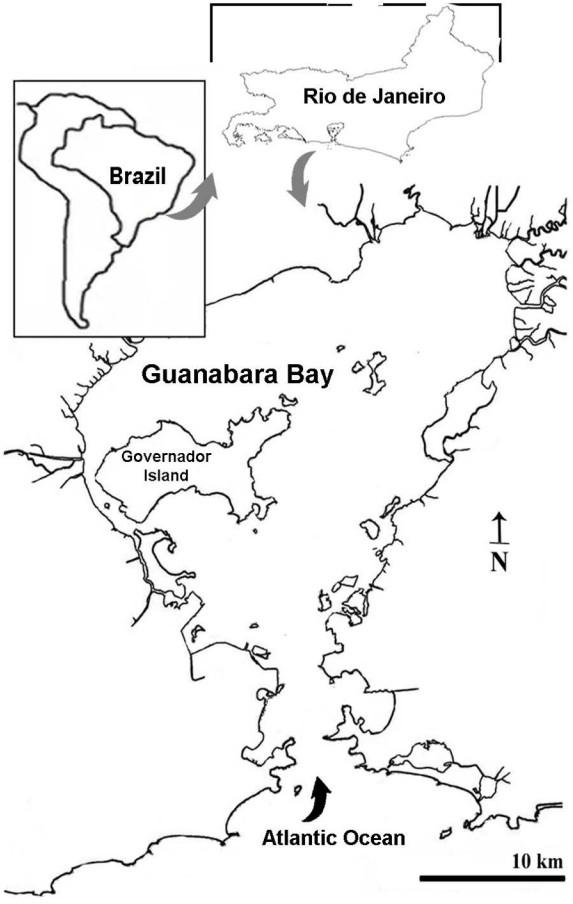

**Fig 1. Geographical location of the Magé mangrove in Rio de Janeiro, Brazil.** Sampling site is indicated in the map by the green ellipse.

of the stock solution (40 mg L⁻¹) using a mineral salt medium (MSM). This poor nutrient media allows DEHP to be the sole carbon source in the medium. Concentrations were chosen based on environmentally relevant concentrations and toxicity tests using aquatic species (e.g., $EC_{50,48h}$ *Daphnia magna*: 0.16 mg L⁻¹ - ECOTOX Database; $EC_{50,72h}$ microalga *Pseudokirchneriella subcapitata*: 0.003 mg L⁻¹ - OECD Test Guideline 201).

The treatments were performed in three replicates of 125 mL Erlenmeyers containing 1 mL of the bacterial inoculum (approximately 2.7 x10⁹ cells mL⁻¹) and 14 mL of exposure concentrations: 0.05; 0.09; 0.19; 0.38; 0.75; 1.50; 3.00 and 6.00 mg DEHP L⁻¹. Three negative control replicates were prepared with MSM and the bacteria inoculum, and a blank control was prepared with MSM bacterial-free. The experiment was carried out in the incubator (SolidSteel) at 35°C ± 1, and aliquots (1 mL) of treatments were taken at 24, 48, 72, and 96 h. The aliquot absorbance (600 nm) was measured using a Trilogy Laboratory Fluorometer (Turner Designs) to detect changes in bacterial biomass.

## Identification and characterization of DEHP-resistant microorganisms

A subsequent laboratory experiment was performed to identify the composition of the consortium MP001 and characterize the microorganisms resistant and with the potential to degrade the DEHP. A volume of 1 mL of the MP001 consortium (approximately 2.7 x10⁹ cells mL⁻¹) was added to 125 mL Erlenmeyers with 10 mL of MSM spiked with 0.38 mg L⁻¹ of DEHP

as the sole carbon source. DEHP concentration was chosen based on the results of the DEHP resistance assays (section 2.4). Additionally, Erlenmeyers only with MSM and DEHP served as a negative control for microorganisms. The experiment was performed in duplicate and carried out in the incubator (SolidSteel) at 35°C ± 1 for 48 h. After 48 h, the total volume of the Erlenmeyers was centrifuged at 4,300 rpm for 15 min at room temperature, washed once with 0,9% saline buffer, and placed in 15 mL Falcons for further high-throughput sequencing.

## Sample sequencing and bioinformatic

For DNA sequencing, 25 mL of the culture medium was centrifuged at 4,000g. The pellets were used as a source of total DNA. The extraction was performed using the DNeasy PowerSoil kit (Qiagen®) following the manufacturer's recommendations. Qualitative verification of the extracted DNA was conducted by agarose gel electrophoresis (1%) (Thermo Fisher Scientific™). Libraries were constructed using the Illumina 16S Metagenomic Sequencing Library Preparation protocol (Illumina, San Diego, CA, USA). Amplification of the V3 and V4 regions of the 16S ribosomal gene was achieved through polymerase chain reaction (PCR) for bacterial and archaeal identification, using the universal primer pairs S-D-Bact-0341-b-S-17 and S-D-Bact-0785-a-A-21 [48]. Amplicon fragment sizes were assessed by capillary electrophoresis using Agilent 4200 TapeStation (Agilent Technologies, Santa Clara, CA, USA) to ensure quality. Subsequently, samples were purified using the Agencourt AMPure XP Kit (Beckman Coulter, Inc., Brea, USA), following the manufacturer's instructions. Indexes were then added to each sample through PCR Indexing using the Nextera XT Library Preparation Kit indexes (Illumina, San Diego, CA, USA). Afterward, samples were purified and quantified as described above. Libraries were standardized to a concentration of 2 nmol L$^{-1}$ for genomic pool preparation following the 16S metagenomic sequencing library preparation protocol (Illumina, San Diego, CA, USA). Paired-end sequencing was performed on the Illumina NexSeq 2000 platform using the NextSeq 1000/2000 P2 Reagent Kit (600 Cycles) sequencing kit.

Bacterial identification employed the PIMBA pipeline [49], a pipeline based on the QIIME (Quantitative Insights Into Microbial Ecology) pipeline (Caporaso et al., 2010). Initially, sequences were trimmed and quality-filtered (Phred >20) using Prinseq [50]. Subsequently, sequences were assembled using the Pear assembler [51]. To improve the quality of the metabarcoding, all sequences shorter than 100 bp were filtered and sequences with >97% similarity were grouped into Amplicon Sequence Variants (ASV) using Swarm 2 [52]. The taxonomy of the ASV was determined by comparing them with sequences available in SILVA132 database [53].

Potential physiology analyses were conducted using 16S rDNA sequencing data. ASV relative abundance and taxonomy data were fed into the FAPROTAX 1.2.6 - Functional Annotation of Prokaryotic software [54]. The FAPROTAX application converts taxonomic profiles of the microbial community into functional profiles, providing information about the microbial community's different metabolic stages, both active and latent. Thus, it does not assess the actual function at the time of sampling, but rather the potential function. All analyses and graphs were generated using the R software version 4.1.2 [55] with the Phyloseq package [56].

## Statistic

Generalized Linear Model (GLM) was applied to the absorbance data obtained in the DEHP resistance assays to test the effects of concentration, time, and concentration*time interaction. When a significant result was obtained in the GLM, the Tukey test was applied *a posteriori*. Data was previously tested for the parametric assumptions - normality and homogeneity of variances - using the Kolmogorov-Smirnov and Levene tests, respectively, and the analyses were performed using the software Statistica 10 (StatSoft).

A Kruskall-Wallis test was performed using the R software version 4.1.2 to measure the differences between the diversity indices of the MP001 inoculum, the DEHP treatment, and the control, and determine the influence of DEHP in the consortium MP001. The Statistical significance was determined by p-value < 0.05.

## Results

### DEHP resistance of bacterial consortium MP001

The *in vitro* response of the MP001 consortium revealed a significant tendency of bacterial biomass increase concerning the negative control, throughout short-term incubation with DEHP (Fig 2). Increases in bacterial biomass induced by DEHP, in proportion to negative control, were more pronounced after 48–96h of incubation showing mean values higher than control (Fig 2). A significant time (GLM, $F_{3,72}$= 15.07, p < 0.0001) and DEHP concentration (GLM, $F_{8,72}$= 11.46, p < 0.0001) effect was found on bacterial biomass. However, no significant effect of the interaction time*concentration was detected on bacterial biomass (GLM, p= 0.09)

### Characterization of consortium MP001

Pronounced increases in bacterial biomass were detected after 48h of incubation concerning the negative control. Thus, the concentration of 0.38 mg L$^{-1}$ was chosen to be used in the further experiment to characterize changes in the

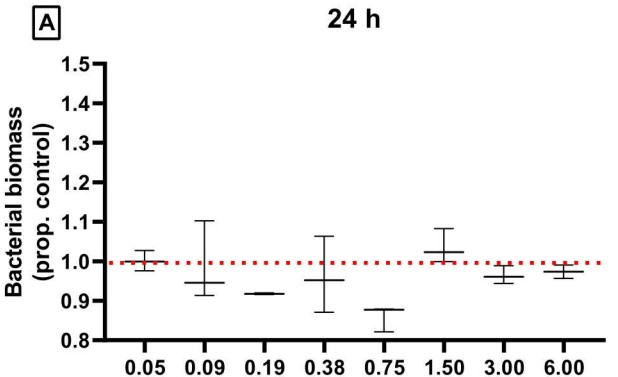
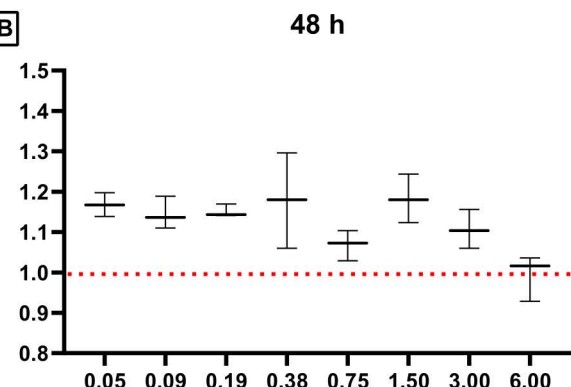
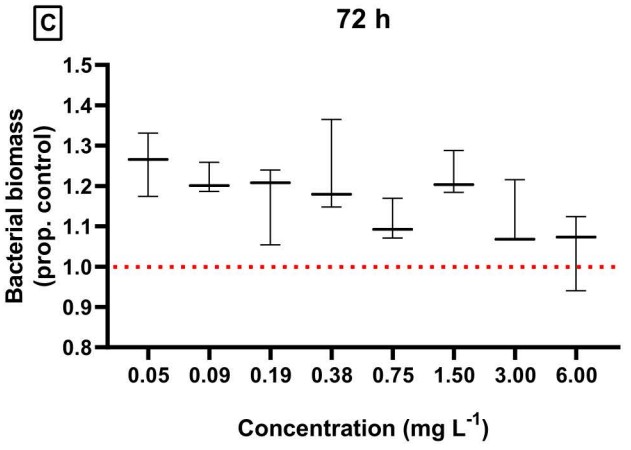
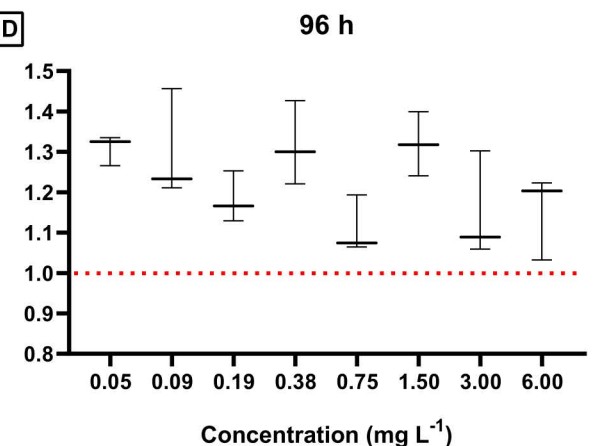

**Fig 2. Variations in bacterial biomass from the consortium MP001 exposed to eight different concentrations of DEHP at 24, 48, 72, and 96h (A, B, C, and D, respectively).** Data are shown as mean values of absorbance in independent replicates (*n*= 3) corrected by the absorbance in negative controls ± standard deviation. The red dashed lines indicate 100% compatibility between the absorbance measured in aliquots obtained in treatments with DEHP and negative controls.

diversity, ecological function, and composition of the consortium MP001 after exposure to DEHP using high-throughput sequencing.

**Composition of the consortium MP001 and prospection of microorganisms to DEHP degradation.** The same sequencing accomplished to characterize the consortium MP001 was used to elucidate its composition, before and after DEHP exposure, and ASVs were classified at height level up to species. In the inoculum of the MP001 consortium, the phylum Firmicutes was predominant (Fig 3A). *Paraclostridium* sp. was the most abundant species (78.99%), and *Bacillus* sp. was also observed (10.73%) in the consortium MP001 inoculum (Fig 3B). In the negative control, *Paraclostridium* sp. (54.02%), *Pseudomonas stutzeri* (19.44%), and *Staphylococcus* sp. (11.97%) were the dominant species and in the DEHP treatment were found the species *Paraclostridium* (50.00%), *Staphylococcus* sp. (12.72%), *Staphylococcus epidermidis* (10.40%) and *Bacillus* sp. (17.63%). Although not significant, the relative abundance of *Pseudomonas stutzeri* and *Paraclostridium* sp. had decreased (19.44% and 4.02%, respectively) whereas that of *Bacillus* sp., *Staphylococcus epidermidis,* and *Staphylococcus* sp. increased (17.63%, 10.40%, and 0.75%, respectively) in the treatment with DEHP compared to the negative control (Fig 3B).

**Diversity of the consortium MP001 before and after DEHP exposure.** The alpha diversity metrics Chao 1 and Shannon indices were used to assess richness and evenness within and between the samples from the MP001 inoculum, the DEHP treatment, and the negative control (Fig 4). The consortium MP001 diversity represented by the inoculum was not substantial (Kruskall-Wallis; $p > 0.05$). In addition, the difference between the diversity of microorganisms in the treatment with DEHP and the negative control was not significant (Kruskall-Wallis; $p > 0.05$)

**The potential ecological function of the consortium MP001 before and after DEHP exposure.** The cluster analyses obtained by the ward.D2 method were performed to assess the ecological functions of the consortium MP001,

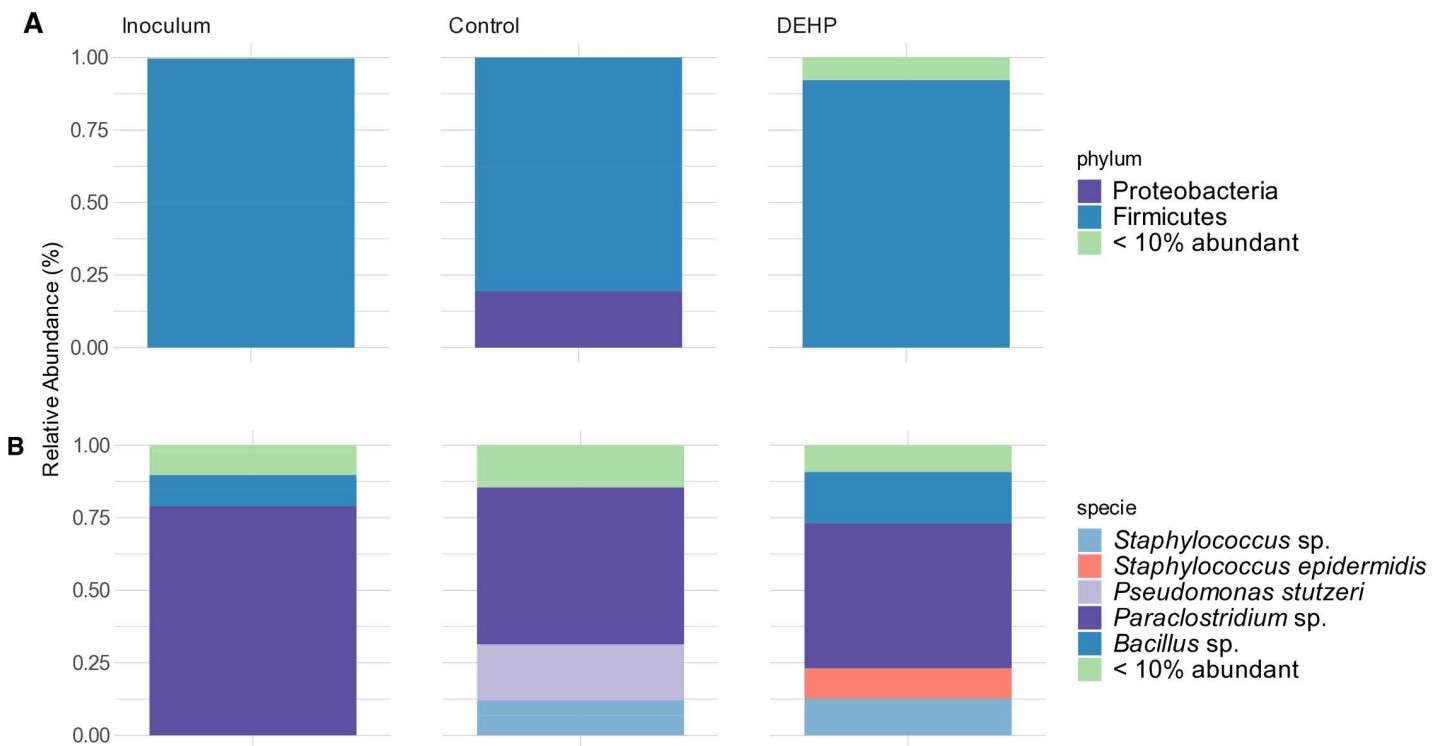

**Fig 3. Bacterial composition of consortium MP001 at the phylum (A) and species (B) level in the MP001 inoculum, negative control, and DEHP treatment after 48 h of DEHP exposure.**

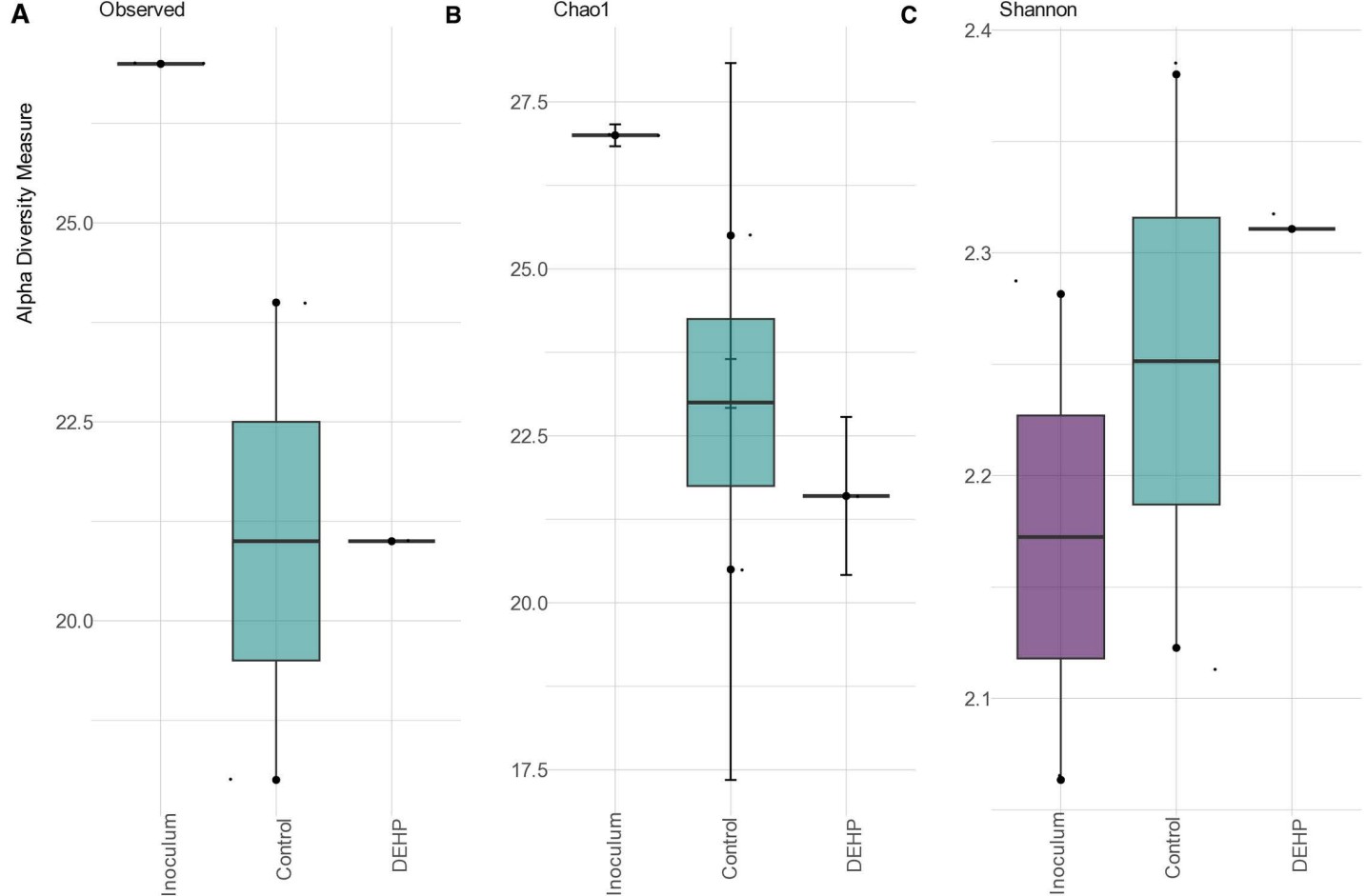

**Fig 4. Alpha diversity of the MP001 inoculum, negative control, and DEHP treatment observed (A) and measured by the Chao 1 (B), Shannon (C), and Simpson (D) indices after 48 h of DEHP exposure.**

and the differences between microorganisms from the MP001 inoculum, DEHP treatment, and negative control (Fig 5). Basically, the principal element used by the bacterial consortium was nitrogen. The main potential ecological function observed was fermentation, which is one of the two ways that chemoheterotrophy is responsibly done through energy production. The potential ecological functions were more intense in the DEHP treatment, and an increase of approximately 30% and 10% was observed, respectively, from the inoculum and from the negative control to the DEHP treatment (Fig 5).

## Discussion

### DEHP resistance of bacterial consortium MP001

Anthropogenic activities have significantly impacted coastal and marine environments. Urbanization, shipping, industrial, and petrochemical activities contribute to high concentrations of pollutants in marine sediments [14]. This is also true for Brazil; oil spills, plastic debris, and untreated sewage discharges have historically affected Rio de Janeiro's coast and Guanabara Bay, significant sources of phthalate DEHP for the sediments. In previously published studies, DEHP concentrations were noted at $2.7 \times 10^{-5}$ mg g$^{-1}$ at the bottom of Guanabara Bay, where the Magé mangrove is located, and $6.90 \times 10^{-5}$ mg g$^{-1}$ near Governador land [57]. In the water, concentrations were found at $2.83 \times 10^{-5}$ mg g$^{-1}$ (Santos

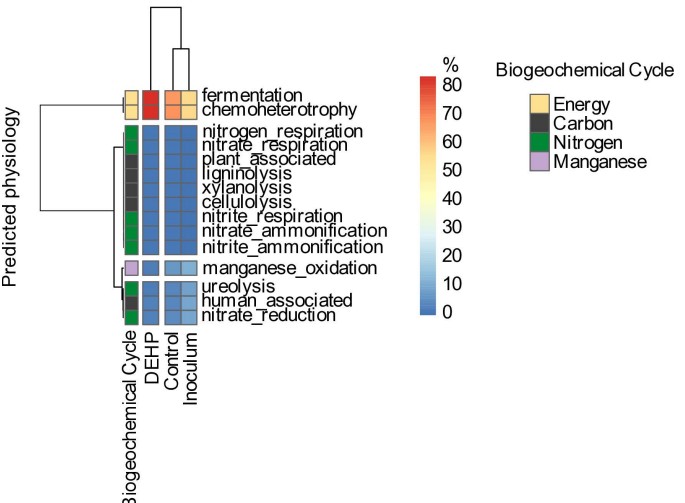

**Fig 5. Potential ecological functions of the MP001 consortium inoculum, negative control, and DEHP treatment after 48 h of DEHP exposure.** The percentages are the relative contribution of the consortium to the biogeochemical cycle and ecological functions and are indicated by the gradient of colors.

Dumont Airport, in downtown Rio de Janeiro) and $3.8 \times 10^{-5}$ mg g$^{-1}$ (in the center of the bay, near the Rio-Niterói bridge) [57]. These DEHP concentrations are considered small compared to other areas where industrialization occurred faster. For example, in Korea, the concentrations ranged from $3.6 \times 10^{-3}$ to $8.3$ mg L$^{-1}$ (sediment Lake) [58]. In coastal areas from the Persian Gulf, DEHP ranged from $1.99 \times 10^{-3}$ to $1.0 \times 10^{-1}$ mg g$^{-1}$ in the sediment [59] and from $5.71$ to $18.51 \times 10^{-3}$ mg L$^{-1}$ in the seawater [60]. In the Mediterranean Sea, DEHP was observed from non-detected values to $1.68 \times 10^{-1}$ mg L$^{-1}$ [61]. Although lower, this is a warning that DEHP is ubiquitous in industrialized areas, and bioremediation processes and searches for bioremediation tools should be encouraged.

The MP001 consortium is an interesting candidate to be used in the phthalate DEHP bioremediation. It was exposed to DEHP and grew using this toxic compound as a carbon and energy source for biomass increase in short-term incubations, independently of toxicant concentration. From 48 to 96 h of incubations, in all the concentrations, the bacterial biomass was higher in the consortium exposed to DEHP than in negative controls. Finding consortiums able to resist and degrade DEHP can be critical to eradicating this compound and its harmful environmental effects [41,62,63]. DEHP is a pollutant that induces disruptive endocrine effects and is already ubiquitous in aquatic systems worldwide, highlighting the importance of biological approaches to eliminate this compound from the environment [4]. The bioremediation using bacterial consortium was demonstrated to be greatly efficient for contaminant removal [40,41,62,63]. Thus, the results obtained here highlight that a bacterial consortium (MP001) isolated from a neotropical mangrove has the potential to be an eco-friendly alternative to DEHP bioremediation, reinforcing the need for further complementary studies using this inoculum to confirm its usefulness for DEHP remediation.

Although the DEHP concentrations after the consortium MP001 exposure and biodegradation rates should be evaluated, the resistance and growth of bacteria in the consortium seem to indicate the consortium's degradation process during the short-term incubation. It is important to highlight that no other source of carbon and energy besides DEHP was added to the mineral medium with the inoculum, and all analyses were performed using the initial inoculum and/or the negative control (without the contaminant) for comparisons. A previous study demonstrated that another consortium, isolated from activated sludge, was able to degrade more than 93.84% of 1,000 mg L$^{-1}$ DEHP in 48 h [62]. The CM9 consortium only in 24 h could degrade 76.89%, 87.86%, 92.82%, 98.25%, and 98.34% of 5, 10, 20, 50, 100, and 200 mg L$^{-1}$ of

DEHP, respectively [41]. Posteriorly, the CM9 consortium completely degraded the DEHP in 72 h of incubation [41]. Similarly, in 3 days, the consortium An6, composed of *Gordonia* sp. and *Pseudomonas* sp., degraded 97.65% of 500 mg L$^{-1}$ DEHP [63]. The degradation rate of the *Rhodococcus pyridinivorans* XB, a facultative anaerobic strain isolated from active sludge, was estimated at 98.95% within 48 h [64]. The strain *Gordonia terrae* RL-JC02 also could completely degrade 50 mg L$^{-1}$ of DEHP within three days [38]. Thus, these significant degradation rates in periods similar to the incubations performed with the consortium MP001 suggest that short-term incubations efficiently demonstrate the ability of microorganism consortiums to resist DEHP exposure and degrade. Furthermore, these findings reinforce that the MP001 consortium could use DEHP as a growth substrate and, perhaps, degrade it. Indeed, a bacterial community from a mangrove rhizosphere has already been found to accelerate DEHP degradation [44].

The most studied bacterial resistance is antibiotic resistance (AR) because of the global health problem of antimicrobial resistance genes (ARGs) [65]. However, in the environment, bacteria are exposed to several chemical pollutants and could develop mechanisms to tolerate not only one pollutant, but also a mixture of them [66]. This resistance process would be driven by co-selection, which is known as the selection and expression of two or more genes, even when exposed to only one selective trigger or stressor [67–68]. In the present study, the MP001 consortium was able to resist DEHP exposure at different concentrations, including environmentally relevant ones (i.e., that had already been detected in aquatic systems). Thus, we hypothesized that the MP001 resistance could be the outcome of the mentioned co-selection mechanisms for bacterial resistance given that Firmicutes, the phylum predominant in the MP001 consortium, is one of the major phylums responsible for dissipation, maintenance, and propagation of ARGs [68]. In addition, bacteria carrying ARGs have already been found in Guanabara Bay [69], the bay that surrounds the mangrove where the MP001 consortium was isolated, which is a source of chemical contamination to the mangrove [70], which probably influences their bacterial composition.

### Characterization of consortium MP001

**Composition of the consortium MP001 and prospection of microorganisms to DEHP degradation.** Regarding the bacterial composition, Firmicutes were the phylum more abundant in the MP001 consortium and it also increased in DEHP treatment compared to negative control, although Firmicutes are rarely found as a dominant group in natural samples [71]. Similarly, Firmicutes were the dominant group in the most polluted site among sediment samples from a China river [72]. In contaminated soils, Firmicutes biomass was found to increase in treatments with 10 and 40 mg L$^{-1}$ of DEHP sediment when compared to control [73]. Microbial composition can be affected by the presence of contaminants [44,72,74,75] and the composition of indigenous bacterial communities can be simplified after PAE exposure [30]. Thus, in agreement with previous studies, the increase in the relative abundance of the Firmicutes group from the MP001 consortium when exposed to DEHP seemed to be induced by the contaminant, suggesting this group was highly tolerant to this PAE and an opportunist bacterial group in environments contaminated with DEHP.

Proteobacteria is one of the most dominant phylum in wetland sediments [43]. Furthermore, an increase in Proteobacteria was reported in experimental assays with DEHP presence [41,76]. Contrarily, in the present study, Proteobacteria was slightly found in MP001 inoculum (i.e., initial microorganisms) and decreased in the DEHP treatment, despite being present in the negative control. Proteobacteria include chemical oxygen demand (COD) bacteria [76], which may explain the low presence of this group in a consortium isolated from an environment with anoxic conditions, such as mangrove sediment [43]. In addition, the fermentative function was detected in bacteria from DEHP treatment, a degradation process played by anaerobic bacteria, which is not found in the Proteobacteria group.

The most abundant genus found in consortium MP001 inoculum, DEHP treatment, and negative control was *Paraclostridium*. *Paraclostridium* is an obligate anaerobe, gram-positive bacteria producer of endospores belonging to the phylum Firmicutes, class Clostridia, and family Peptostreptococcaceae [77]. The family Peptostreptococcaceae is distributed from humans to several environmental habitats, including oil mills, gut microbiota, sludge, contaminated water, marine

sediment, feces, sugarcane bagasse, and fermented food [77–79]. The genus *Paraclostridium* has two species described: *P. benzoelyticum* and *P. bifermentans. Paraclostridium bifermentans* is often associated with infectious diseases; however, these bacteria are rare human pathogens and can have other roles [77]. *Paraclostridium* was reported as able to degrade environmental pollutants [80–83]. For example, the removal of the antibiotic Ciprofloxacin (CIP) was increased by *Paraclostridium* sp. strain S2, when the genus was used as bioaugmentation in a bioreactor [81]. Because the mangroves from where MP001 was isolated receive contamination from the surrounding land and Guanabara Bay waters [70], it is possible that indigenous microorganisms from that environment were under selective pressure and developed strategies to resist and degrade certain pollutants. Therefore, due to its resistance to pollutants, *Paraclostridium* might be the most abundant genus in the consortium MP001.

The genus *Bacillus* was also found in the MP001 consortium, and its relative abundance increased in DEHP treatment compared to the negative control. Similarly, DEHP exposure has been reported to increase the relative abundance of *Bacillus* members found in bacterial consortiums [30]. *Bacillus* species have been reported as biofilms producers [84–85]. The biofilm formation occurs in response to the operon genes *SurfA*. This gene is essential for surfactin synthase, a signaling molecule that triggers Quorum sensing (QS) and stimulates responses of subpopulations to environmental stress, including biofilm formation [85]. Biofilms are composed of microbial cells associated with a self-produced extracellular matrix that provides resistance to biotic factors and chemical pollutants [68]. This strategy is considered one of the co-screening mechanisms for bacterial resistance, which is established by the improvement of the media for bacterial signal transduction and genetic exchange [66,68,86]. Due to the *Bacillus* presence in the MP001 consortium, this shielding might be one of the reasons for the MP001 growth and resistance to DEHP.

The *Bacillus* growing in DEHP treatment could also indicate a capacity for DEHP degradation. *Bacillus* was the most abundant genus on 4th day of DEHP degradation by a marine sediment consortium [87]. In another study, the three better strains capable of DEHP degradation in mangrove sediment belonged to the genera *Bacillus* [88]. It was also suggested that the *Bacillus* members responsible for DEHP degradation were facultative anaerobes [89]. Moreover, the esterase gene encoding enzymes involved in DEHP degradation was promoted in the microbial community of anaerobic soil contaminated with DEHP [90]. Since mangroves can be an anaerobic environment [43], it is possible that the degradation could be accomplished by facultative anaerobic *Bacillus* in the MP001 consortium.

The *Staphylococcus epidermidis* and *Staphylococcus* sp. also increased in DEHP treatment compared to the negative control. The genus *Staphylococcus* is well-known as an infectious bacterium [91]. Specifically, *Staphylococcus epidermidis* is a colonizer of human skin and the main responsible for nosocomial infections [92]. However, it is suggested that the molecular determinants that promote evasion by *S. epidermidis* lead it to cause disease to have original functions in the non-infectious lifestyle of this bacteria [92]. *S. epidermidis* is a biofilm producer with specific proteins to adhere to surfaces, such as MSCrAMMs (microbial surface components recognizing adhesive matrix molecules), and also synthesize a poly-N-acetylglucosamine (PNAG) that surrounds and connects *S. epidermidis* cells in a biofilm [92]. It also bears genes that promote protection against adverse environmental conditions [92]. Moreover, *S. epidermidis* was reported as able to remove the contaminant triphenylmethane dyes [93]. Indeed, some bacteria from the genus *Staphylococcus* can degrade environmental contaminants, such as surfactants, pesticides, and mainly polycyclic aromatic hydrocarbons (PAHs) [94–99]. For instance, *Staphylococcus* sp. strain DAB-1W could degrade the insecticide lindane, an organochlorine compound, at a rate of 15% and 98% in 2 and 8 days, respectively [100]. The strain *S. haemoliticus* 10SBZ1A degraded in saline conditions about 80% of 20 μmol/L of benzo[a]pyrene (BaP) in 25 days [98]. The degradation of PAH with high molecular weight, for example, the BaP, may indicate the capability of *Staphylococcus* to degrade DEHP, a PAE with a longer alkaline chain [36,38] Furthermore, the degradation in saline conditions was achieved because *Staphylococcus*, including *S. epidermidis* [92], is a halotolerant bacterium [91]. This salt-tolerant characteristic could benefit the presence of this genera in the mangrove ecosystem that receives marine water and, hence, in the MP001 consortium.

The degradation capability of *Staphylococcus* and *Staphylococcus epidermidis* might also be influenced by its ability to resist chemical exposure obtained from resistance genes. *Staphylococcus* bears mobile genetic elements, including the staphylococcal cassette chromosome mec (*SCCmec*), a mobile genetic element from the genera [101]. *SCCmec* carries the *mec* gene (*mecA*, *mecB*, and *mecC*) and the genes that control their expression. Besides, it is formed by three regions: the *ccr* gene complex, the *mec* gene complex, and the joining region (J region). In the *ccr* gene complex can be inserted multiple antibiotics and heavy metal-resistant genes [101]. It is also integrated into the chromosome of *Staphylococcus* strains, making possible the change of genetic information between *Staphylococcus* strains to adapt to the environmental stress and the pressure of antibiotics [101]. *Staphylococcus epidermidis* carries the SCCmec, which holds ten different SCCmec structures and is a great reservoir of antibiotic-resistance genes [92]. Therefore, it is likely that other resistant genes of *Staphylococcus* were inserted in the *ccr* gene complex and conferred resistance to other pollutants, such as DEHP.

A decrease in *Pseudomonas stutzeri* was also observed in the DEHP treatment compared to the negative control. *Pseudomonas* is a group already found capable of degrading pollutants [102–103], and *Pseudomonas* strains were found to degrade DEHP [35,63]. Three strains of this group — *Pseudomonas* sp. PKDM2, *Pseudomonas* sp. PKDE1, and *Pseudomonas* sp. PKDE2 —degraded 500 mg L$^{-1}$ of DEHP [35]. In contrast, the strain *P. fluoresences* FS1 degraded less than 20% of 100 mg L$^{-1}$ DEHP in 3 days [36]. Nevertheless, all these *Pseudomonas* sp. strains degrade more efficiently short-chain PAEs than longer-chains, such as DEHP [35, 36]. Despite that, DEHP was highly degraded (97.65%) when a *Pseudomonas* strain — *P. putida* ShA — was in a consortium with other bacteria efficient in degrading long-chain PAEs [63]. Some microorganisms can not tolerate the toxicity of some pollutants to persist alone, but they can when the compound is metabolized to a less toxic intermediate by a syntrophic microorganism [39]. Besides, they can use the metabolites as substrate resulting in the total mineralization of the pollutant [39]. This highlights the importance of consortiums in the degradation process of chemical contaminants. Indeed, *Pseudomonas* have been found in consortiums efficient in pollutant-degrading [104–107], including together with *Staphylococcus*, the other genera also found in the MP001 consortium [108]. Therefore, the *P. stutzeri* decrease in the MP001 consortium could indicate their low resistance to DEHP, but with the other bacteria in the consortium, they could collaborate in the biodegradation process.

P . *stutzeri* is a cosmopolitan bacterium with a relevant role in nitrogen cycling [109]. It has high physiological and genetic adaptability, which could be explained by chemotaxis, genetic mobile elements, and competence [109–110]. Besides, this bacterium is chemeoheterotrophic and can grow in minimal media with a single carbon source [110], similar to the present study. *P. stutzeri* can also degrade xenobiotic compounds, including biocides [109, 111] and several aromatic hydrocarbons, such as phenanthrene [112–114], naphthalene [115–116], butylbenzene [117], pyrene [113–114], benzanthracene [113], petroleum hydrocarbons [118–119], among others. Regarding PAEs, *P. stutzeri* has been found capable of degrading 5 mg L$^{-1}$ of DBP with a half-life of 1.8 days, and likely to our study, could not survive after 5 days of incubation with other microorganisms of the microbial community from their sampling site [120].

**Diversity of the consortium MP001 before and after DEHP exposure.** A low alpha diversity was found in the MP001 consortium with DEHP exposure. Reduced diversity in the DEHP presence has been already reported. Bacterial consortiums enriched with DEHP demonstrated lower diversity than environmental samples not exposed to DEHP [87]. In a microbial soil community, despite being little restored, the diversity had a strong decrease within 7 days of DEHP exposure [41]. A negative effect on the richness, evenness, and Shannon diversity was observed in an anaerobic bacterial community from soil exposed to DEHP [89]. In addition, the richness of microorganisms was decreased with the DEHP addition in activated sludge of landfill leachate [76]. Thus, DEHP seems to be a compound able to decrease the diversity of microbial communities.

However, we point out that not only in the DEHP treatment (compared to the negative control) that a low diversity was found, but also in the negative control and MP001 inoculum. Microbial communities are highly diverse, yet the diversity

can be smaller in stressed environments, such as polluted ones [121]. A significantly lower diversity index of bacterial community was reported in sediments of heavy black-odors rivers with higher concentrations of contaminants (PAHs and PAEs, including DEHP) than in moderate ones with less concentration of contaminants [122]. In sediment samples from estuaries, the OTU/ASV richness and Shannon index were negatively influenced by the contamination with the heavy metal copper [75]. Similarly, in a riverine microbial community, the alpha diversity was negatively correlated with the total concentration of pharmaceutic and personal care products (PPCPs) [74]. Since mangroves can intercept a large amount of contamination due to their geochemistry [13,15] and consortium MP001 was isolated from a mangrove surrounded by the highly polluted Guanabara Bay [69–70], this in situ pollution might have decreased the diversity of their bacterial community and, consequently, of the consortium MP001.

Moreover, we can make some important inferences about the lack of significant change in community diversity after DEHP treatment. The main one is how resistant the bacteria observed in the environment are to the DEHP. Organisms have already been chronically exposed to the DEHP and have been selected in nature [14,57]. Thus, it is possible that the selection that occurred in the Magé mangrove resulted in resistant bacteria, thus, decreasing their diversity. The low alpha diversity and metabolic pathways might also be influenced by the experimental conditions. However, in the present study, we aimed to assess and characterize a biothecnological tool to DEHP biorremediation and not specifically assess the diversity of the local microbiome.

**The potential ecological function of the consortium MP001 before and after DEHP exposure.** The consortium MP001 presented low diversity in terms of potential metabolic pathways compared to negative control, with only one energy production function (fermentation) and several pathways that represent the reworking of nitrogen compounds (e.g., pathways linked to ammonification and nitrate reduction). Due to the grouping, the functions between the inoculum and the control are very similar, and potential metabolic functions become different with exposure to DEHP, where the consortium made a significant investment in the fermentation process to grant more energy.

Fermentation is the primary mode of organic matter degradation. Microorganisms influence the functioning of ecosystems by mediating biogeochemical cycles [72]. The fermentative bacteria hydrolyze the organic compounds and ferment the products generating $CO_2$, acetate, and $H_2$, which are substrates for anaerobic respiration carried out by the methanogenic bacteria [43,123]. Fermentation is carried out by obligate or facultative anaerobic bacteria [43], which agrees with those found in the present study such as *Bacillus* and *Paraclostidium*. *Paraclostidium*, the more abundant genus in the MP001 consortium, has already been found as fermentative bacteria [78]. The strain *Paraclostridium* sp. CR4 was reported as producing $H_2$, mainly from glucose, and being able to be used in fermentative reactors [78]. Furthermore, Firmicutes was the more abundant phylum in the MP001 consortium. Similarly, this group was dominant in the microbial community of sugar cane, degrading the sugar by fermentation [71]. Thus, the fermentation function in the MP001 consortium indicates the possibility of DEHP degradation via fermentation. Furthermore, since wetlands can be an anoxic environment and fermentation occurs in these ecosystems [43], this may be the reason that fermentation has been found in the inoculum of the MP001 consortium.

The chemoheterotrophy and low carbon and nitrogen function have been also observed. Organic pollutants can influence the biogeochemical cycles' ecological functions by altering microbial composition, activities, or gene expression levels [122]. The relative abundance of nitrogen and phosphorus-related genes was lower in highly contaminated sediments than in moderate ones, suggesting a decrease in nitrogen and phosphorus metabolism [122]. Similarly, the abundance of functional genes in bacterial communities from river sediment samples was negatively correlated with the different pollution levels from each one. It was observed that the most polluted river had a lower abundance of functional genes, including energy metabolism-related, such as nitrogen metabolism, sulfur metabolism, and photosynthesis [73]. Thus, a possible reason for the low percentage of carbon and nitrogen function in the MP001 consortium is the pollution from their source.

## Conclusion

A consortium isolated from a neotropical mangrove (MP001) was characterized to assess its resistance to DEHP, composition, diversity, and potential ecological function. The MP001 consortium composition was dominated by the *Paraclostridium* sp., followed by *Bacillus* sp. After 48 h of DEHP exposure, *Paraclostridium* sp., *Pseudomonas stutzeri*, and *Staphylococcus* sp. were found in the negative control, whereas *Paraclostridium* sp., *Staphylococcus* sp., *Staphylococcus epidermidis*, and *Bacillus* sp. in the DEHP treatment. Thus, the relative abundance of *Pseudomonas stutzeri* and *Paraclostridium* sp. had decreased whereas that of *Bacillus* sp., *Staphylococcus epidermidis,* and *Staphylococcus* sp. increased in the treatment with DEHP compared to the negative control, indicating a possible effect of DEHP on the MP001 consortium composition. The alpha diversity of the MP001 consortium was not substantial and no significant difference was found between the negative control and DEHP treatment. Moreover, the main potential ecological function in MP001 inoculum, negative control, and DEHP treatment was fermentation. Notably, the MP001 consortium demonstrates a significant tendency to increase the bacterial biomass concerning the negative control after 48 h of exposure to DEHP. To the best of your knowledge, this is the first study to characterize a bacterial consortium from the Magé mangrove, Rio de Janeiro. Furthermore, the results obtained from this study highlight a potential bacterial consortium that could degrade DEHP and be used to remove this compound from DEHP-contaminated sites, decreasing the harmful effects of this compound on the environment.

## Supporting information

**S1 Dataset. Resistance assay values.**
(XLSX)

**S2 Dataset. Raw sequencing data.**
(XLSX)

## Acknowledgments

The authors are grateful to Fernanda Silva dos Santos for the technical assistance.

## Author contributions

**Conceptualization:** Julia de Morais Farias, Raquel A. F. Neves.

**Data curation:** Julia de Morais Farias, Raquel A. F. Neves, Leandro Araujo Argolo.

**Formal analysis:** Julia de Morais Farias, Raquel A. F. Neves.

**Funding acquisition:** Raquel A. F. Neves, Natascha Krepsky.

**Investigation:** Julia de Morais Farias.

**Methodology:** Julia de Morais Farias, Leandro Araujo Argolo.

**Resources:** Leandro Araujo Argolo, José Augusto P. Bitencourt.

**Software:** José Augusto P. Bitencourt.

**Supervision:** Raquel A. F. Neves, Natascha Krepsky, José Augusto P. Bitencourt.

**Validation:** Raquel A. F. Neves, José Augusto P. Bitencourt.

**Visualization:** José Augusto P. Bitencourt.

**Writing – original draft:** Julia de Morais Farias.

**Writing – review & editing:** Julia de Morais Farias, Raquel A. F. Neves, Natascha Krepsky, José Augusto P. Bitencourt.

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
