## [Decision Letter · Decision Letter 0]

26 Nov 2024

PONE-D-24-45060Mangrove consortium resistant to the emerging contaminant DEHP: Composition, diversity, and ecological function of bacteriaPLOS ONE

Dear Dr. Natascha Krepsky,

Thank you for submitting your manuscript to PLOS ONE. After careful consideration, we feel that it has merit but does not fully meet PLOS ONE’s publication criteria as it currently stands. Therefore, we invite you to submit a revised version of the manuscript that addresses the points raised during the review process.

We look forward to receiving your revised manuscript.

Kind regards,

Mei Li

Academic Editor

PLOS ONE

Journal Requirements:

“Authors are grateful for the research grants attributed by the Foundation Carlos Chagas Filho Research Support of the State of Rio de Janeiro (FAPERJ) to Raquel A. F. Neves (E-26/201.283/2021; E-26/210.024/2024), to Natascha Krepsky (E-26/211.470/2021) and the Graduate Program in Neotropical Biodiversity (E-26/211.043/2021). as In addition to the research grant attributed by the Brazilian National Council for Scientific and Technological Development (CNPq) to Raquel A. F. Neves (PQ2; 306212/2022-6). This study was financed by the Brazilian National Council for Scientific and Technological Development (CNPq), through the scholarship (master’s degree) to Julia de Morais Farias”

“The authors thank to Fernanda Silva dos Santos for the technical assistance. Authors are grateful for the research grants attributed by the Foundation Carlos Chagas Filho Research Support of the State of Rio de Janeiro (FAPERJ) to Raquel A. F. Neves (E-26/201.283/2021; E-26/210.024/2024), to Natascha Krepsky (E-26/211.470/2021) and the Graduate Program in Neotropical Biodiversity (E-26/211.043/2021). as In addition to the research grant attributed by the Brazilian National Council for Scientific and Technological Development (CNPq) to Raquel A. F. Neves (PQ2; 306212/2022-6). This study was financed by the Brazilian National Council for Scientific and Technological Development (CNPq), through the scholarship (master’s degree) to Julia de Morais Farias.”

“Authors are grateful for the research grants attributed by the Foundation Carlos Chagas Filho Research Support of the State of Rio de Janeiro (FAPERJ) to Raquel A. F. Neves (E-26/201.283/2021; E-26/210.024/2024), to Natascha Krepsky (E-26/211.470/2021) and the Graduate Program in Neotropical Biodiversity (E-26/211.043/2021). as In addition to the research grant attributed by the Brazilian National Council for Scientific and Technological Development (CNPq) to Raquel A. F. Neves (PQ2; 306212/2022-6). This study was financed by the Brazilian National Council for Scientific and Technological Development (CNPq), through the scholarship (master’s degree) to Julia de Morais Farias”

Reviewers' comments:

Reviewer's Responses to Questions

**Comments to the Author**

1. Is the manuscript technically sound, and do the data support the conclusions?

Reviewer #1: Yes

Reviewer #2: Yes

2. Has the statistical analysis been performed appropriately and rigorously? 

Reviewer #1: Yes

Reviewer #2: Yes

3. Have the authors made all data underlying the findings in their manuscript fully available?

Reviewer #1: Yes

Reviewer #2: No

4. Is the manuscript presented in an intelligible fashion and written in standard English?

Reviewer #1: No

Reviewer #2: Yes

5. Review Comments to the Author

Reviewer #1: Comments to the authors:

In this manuscript, MP001 communities were exposed to DEHP at different concentrations under laboratory conditions, and high-throughput sequencing techniques were used to assess bacterial composition, diversity, and potential ecological function. The resistance, composition, diversity and ecological function of MP001 to the emerging pollutant DEHP were evaluated. Studies have shown that the MP001 community shows potential as an eco-friendly bioreactor for DEHP. But there are some problems with this manuscript: First, there are some problems with structure and English grammar. Second, you need to provide detailed information about the laboratory equipment and application. Finally, issues related to community function and community diversity warrant further analysis and consideration of possible environmental influences on community structure and function. The specific comments are provided below.

Specific comments:

1. The section "1. Introduction" can add research hypothesis, deeper research purpose and research significance

2. In section "2.2 Sampling and consortium isolation", it is recommended to draw a geographic map of the sample area.

3. In the section "2.2 Sampling and consortium isolation", to what depth was the sediment collected? Are they randomly selected sediments or are they collected at specific locations?

4. In the section "2.3. Bacterial inoculum preparation", please provide the manufacturing information of each experimental instrument.

5. In the section "2. Materials and Methods", I noticed that you used some applications to analyze the data, please provide their version information.

6. Line 156. There is a grammatical error that suggests adding the indefinite article "a" between "as" and "negative".

7. In the section "3. Results", whether attention was paid to the possible effects of other environmental factors (such as temperature, pH, salinity, etc.) on community structure and function.

8. In "3.2.2. Diversity of the consortium MP001 before and after DEHP exposure", there was no significant change in community diversity after DEHP treatment. But it is worth exploring in depth whether this means that the community's resistance to DEHP is due to its inherent diversity or due to other factors.

9. . In the section "3.2.3. Potential ecological function of the consortium MP001 before and after DEHP exposure", although fermentation is mentioned as the main potential ecological function, Other possible ecological functions or interactions among community members were not further explored.

10. It is suggested to present the "4. Discussion" part in subsections.

Reviewer #2: Comments:

In this study, the authors isolated a bacterial consortium from neotropical mangrove using Di(2-ethylhexyl) phthalate (DEHP) as a stress source. They aimed to identify and characterize the DEHP-resistant microorganisms through gradient concentration exposure, DNA sequencing. The findings demonstrated that MP001 possesses a high potential for application in bioremediation purposes. However, significant concerns exists regarding the environmental significance and reasonability of the exposure design in its current form. Additionally, there are issues with the layout of figures and redundant/inaccurate information conveyed in some figures, which greatly compromises the readability of this paper.

The following specific points require further clarification and addressing:

1.In the Abstract, the authors mentioned the bacterial composition of exposure treatment and negative control. Are there any conclusion based on the changes in treatment and negative control groups?

2.The background and environmental context in which the study was conducted lack clarity. The selection of exposure doses ranged from 0.05 to 6 mg/L was based on specific considerations. However, there is a lack of available data or descriptions regarding the actual range of DEHP concentrations in the environment, which forms the core focus of this research.

3.In the present study, MP001 was isolated from the sediment of the Magé mangrove incubated at 37 ℃, and the exposure experiment was carried out at 35 ℃ ± 1. However, the average temperature in Rio de Janeiro state was reported as 17-36 ℃. Why did the authors choose such high temperatures? Does it really make sense to conduct bacteria isolation and exposure at these temperature?

4.In this study, the sediment sample was collected from the Magé mangrove in Rio de Janeiro state, which seems that the sediment sample was sampled at only one sampling site in the Magé mangrove. I’m worrying about the representativeness of this study because of the low sample size. If not, please provide the sampling sites of the study.

5.Could the media used for the separation and culture have a certain effect on bacteria determination? Why the media employed to bacteria isolation and exposure experiment were different?

6.The authors demonstrated that DEHP could provide as a carbon resource and degraded by the MP001 consortium in Line 313-314, Page 13. However, DEHP concentrations after exposure were didn’t determine. Thus, I consider this conclusion is unreasonable.

7.In Discussion section, the authors explain the dominant role of ARGs in xenobiotics resistance. However, it seems that the ARGs identification was absent in the present study.Therefore, I consider many elements of these paragraphs to be redundant and irrelevant to the content of this paper.

8.Ensure consistency in the significance in italics across the manuscript. Besides, several mistakes, such as “DEH Psediment” in Line 352, Page 14, disordered graphs (Figure 2 and 3)......

9.The authors should check and correct the mistakes in references according to the format requirement of the journal.

10.Several journal abbreviations need to be fixed and standard journal title abbreviations should be used throughout.

6. PLOS authors have the option to publish the peer review history of their article (what does this mean? ). If published, this will include your full peer review and any attached files.

**Do you want your identity to be public for this peer review?** For information about this choice, including consent withdrawal, please see our Privacy Policy .

Reviewer #1: No

Reviewer #2: No

---

## [Author Response · Author response to Decision Letter 1]

30 Dec 2024

Thank you for sending the manuscript entitled “Mangrove consortium resistant to the emerging contaminant DEHP: Composition, diversity, and ecological function of bacteria” to review. We would like to acknowledge the reviewers' excellent suggestions for improving our manuscript's revised version. The comments made by the reviewers and the responses are shown below. Edits in the manuscript were highlighted in yellow.

Reviewer #1:

1. The section "1. Introduction" can add research hypothesis, deeper research purpose, and research significance

Response: We thank Reviewer 1 for the suggestion. We now included a deeper research purpose and significance in the introduction (Pag 4, Lines 100-106).

2. In section "2.2 Sampling and consortium isolation", it is recommended to draw a geographic map of the sample area.

Response: We acknowledge and thank the request to provide a map of the sample area. We now included the sample area map in the manuscript (Pag 5).

3. In the section "2.2 Sampling and consortium isolation", to what depth was the sediment collected? Are they randomly selected sediments or are they collected at specific locations?

Response: We thank and agree that the depth of sediment collection should be provided. We now add the depth of sediment collection (Pag 5, Line 117). The collection was made randomly in a site known for receiving contamination from an anthropogenically impacted bay. The site was chosen based on the lower cost and accessibility of the sampling and the site's contamination. We changed our text to add the reason for the site chosen (Pag 4, Lines 118-119).

4. In the section "2.3. Bacterial inoculum preparation", please provide the manufacturing information of each experimental instrument.

Response: We thank the reviewer for the suggestion. We now added the manufacturing information of the experimental instruments (Pag 5, Line 135; Pag 6, Line 139 and 156; Pag 7, Line 169).

5. In the section "2. Materials and Methods", I noticed that you used some applications to analyze the data, please provide their version information.

Response: We acknowledge and thank the reviewer for the suggestion. Now, we have included the version information of the statistic program (Pag 8, Line 208; Pag 9, Line 217).

6. Line 156. There is a grammatical error that suggests adding the indefinite article "a" between "as" and "negative".

Response: We thank the reviewer for the careful reading. We now have corrected the sentence (Pag 7, Lines 167-168).

7. In the section "3. Results", whether attention was paid to the possible effects of other environmental factors (such as temperature, pH, salinity, etc.) on community structure and function.

Response: We performed a laboratory experiment. Due to this, we used standardized environmental factors such as temperature and pH to ensure effective supervision and maintain laboratory quality control. From the results with standardized factors, we can do other experiments to assess the influence of temperature, pH, salinity, and other variables in the consortium. However, this was not the focus of this first characterization of consortium MP001. We aimed to assess a biotechnological approach to DEHP bioremediation. Still, we thank the reviewer for raising this question and add in the manuscript the possibility of the influence of the laboratory conditions in the MP001 diversity and metabolic pathways (Pag 21, Lines 526-529).

8. In "3.2.2. Diversity of the consortium MP001 before and after DEHP exposure", there was no significant change in community diversity after DEHP treatment. But it is worth exploring in depth whether this means that the community's resistance to DEHP is due to its inherent diversity or due to other factors.

Response: We thank the reviewer for the careful reading and the suggestion. We made some changes in the discussion to include more details about the community's resistance to DEHP (Pag 21, Lines 521-526).

9. In the section "3.2.3. Potential ecological function of the consortium MP001 before and after DEHP exposure", although fermentation is mentioned as the main potential ecological function, Other possible ecological functions or interactions among community members were not further explored.

Response: Dear reviewer, we made changes in the manuscript to include more details about MP001 physiology (Pag 22, Lines 532-538). Thanks for raising this point.

10. It is suggested to present the "4. Discussion" part in subsections.

Response: We agreed with Reviewer 1 and recognized the need to divide the discussion into subsections. Now the paper discussion is in subsections.

Reviewer #2:

1.In the Abstract, the authors mentioned the bacterial composition of exposure treatment and negative control. Are there any conclusion based on the changes in treatment and negative control groups?

Response: We thank and appreciate the question. In our study, the discussion and conclusions were based entirely on the comparison between the treatments and the negative control results. In the paragraph beginnings, we highlight these comparisons to the readers (Pag 13-14, Lines 332-335; Pag 15, Lines 370-371; Pag 16, Lines 407-408; Pag 17; Lines 429-430; Pag 19, Lines 466-467; Pag 20, Lines 506-507; Pag 22, Lines 532-533; Pag 23, Lines 578-579).

2.The background and environmental context in which the study was conducted lack clarity. The selection of exposure doses ranged from 0.05 to 6 mg/L was based on specific considerations. However, there is a lack of available data or descriptions regarding the actual range of DEHP concentrations in the environment, which forms the core focus of this research.

Response: The raw data we accessed dates back to 2002. In the previous version of this manuscript, we opted not to include it in the text. However, in response to the reviewer's recommendations, we have incorporated this data at the beginning of the discussion, hoping that it will help readers better understand the conditions of the region that provided the sediment (Pag 12-13, Lines 298-314).

3.In the present study, MP001 was isolated from the sediment of the Magé mangrove incubated at 37 ℃, and the exposure experiment was carried out at 35 ℃ ± 1. However, the average temperature in Rio de Janeiro state was reported as 17-36 ℃. Why did the authors choose such high temperatures? Does it really make sense to conduct bacteria isolation and exposure at these temperature?

Response: That’s a good question the reviewer raised. We adhered to strict protocols that guided us throughout the process and allowed us a week to select experimental temperatures near the sampling locations. The temperature used in isolation and experiments must be standardized to ensure effective supervision and maintain laboratory quality control. In the future, when we have a pool of bacteria with biotechnological potential, we will be able to conduct an experiment at a temperature close to that of the study or target region. However, in this manuscript, we choose to highlight the positive results presented by MP001.

4.In this study, the sediment sample was collected from the Magé mangrove in Rio de Janeiro state, which seems that the sediment sample was sampled at only one sampling site in the Magé mangrove. I’m worrying about the representativeness of this study because of the low sample size. If not, please provide the sampling sites of the study.

Response: The main focus of these manuscripts is testing a bacterial consortium that shows a biotechnological potential. It sampled an area with a high environmental impact, as it can be a valuable source for isolating bacteria for bioremediation. It was preferred to perform not to characterize the microbiome of the Magé mangrove. Therefore, we chose to make only one sampling point to identify what microorganisms there are potentially in that area that can survive the isolation process and which are the main candidates that have the potential to survive exposure to DEHP.

5.Could the media used for the separation and culture have a certain effect on bacteria determination? Why the media employed to bacteria isolation and exposure experiment were different?

Response: We thank the reviewer for the attentive question. The media used in the MP001 culture is non-selective. So, it can grow all the bacteria present in the collected sediment and, consequently, allows us to access the bacteria from the environment (our purpose when we looked for a consortium with biotechnological potential). Secondly, the media employed in the bacteria isolation and exposure experiment were different because for the isolation we needed a nutritive media to grow all the bacteria from the environment. Differently, for the exposure experiment, we needed a nutrient-poor media, thus, DEHP would be the only source of carbon and energy to the consortium MP001 and we could access the influence of only DEHP in the decrease or increase of MP001 biomass. To clarify this in the text, we add the reason for the media choose in the experimental assays (Pag 6, Lines 146-147).

6. The authors demonstrated that DEHP could provide as a carbon resource and degraded by the MP001 consortium in Line 313-314, Page 13. However, DEHP concentrations after exposure were didn’t determine. Thus, I consider this conclusion is unreasonable.

Response: We hugely appreciate the reviewer's comment. In Pag 13, we were trying to raise a suggestion that consortium MP001 could be able to degrade DEHP due to their resistance to the compound. However, we understood that the sentence in Lines 313-314 sounds unreasonable. To improve this question, we changed the sentence by adding a doubt adverb (Pag 14, Line 348) and, at the beginning of the paragraph, we included a recommendation for the evaluation of the DEHP concentrations after the consortium MP001 exposure (Pag 13, Line 329).

7.In Discussion section, the authors explain the dominant role of ARGs in xenobiotics resistance. However, it seems that the ARGs identification was absent in the present study.Therefore, I consider many elements of these paragraphs to be redundant and irrelevant to the content of this paper.

Response: We appreciate the request. We aimed to explore the possible reasons for consortium MP001's resistance to DEHP. However, we understand that it may seem redundant. Thus, we removed part of this discussion (Pag 14-15).

8.Ensure consistency in the significance in italics across the manuscript. Besides, several mistakes, such as “DEH Psediment” in Line 352, Page 14, disordered graphs (Figure 2 and 3)......

Response: We thank the reviewer for pointing out the problems in the consistency of italics significance and flaws in the typing. We now corrected the italic (Pag 9, Line 220) and the sentences were rewritten (Pag 15, Line 374). We improve Figures 2, 3, and 4 (now 3, 4, and 5). In Figure 3, we also remove the Simpson diversity index to avoid redundancy.

9.The authors should check and correct the mistakes in references according to the format requirement of the journal.

Response: We acknowledge the request. The references were corrected

10.Several journal abbreviations need to be fixed and standard journal title abbreviations should be used throughout.

Response: We thank the reviewer for pointing out that these abbreviations were wrong. We now corrected them in our manuscript.

Sincerely,

Natascha Krepsky, PhD and Associate Professor at UNIRIO.

---

## [Decision Letter · Decision Letter 1]

21 Feb 2025

Mangrove consortium resistant to the emerging contaminant DEHP: Composition, diversity, and ecological function of bacteria

PONE-D-24-45060R1

Dear Dr. Natascha Krepsky,

We’re pleased to inform you that your manuscript has been judged scientifically suitable for publication and will be formally accepted for publication once it meets all outstanding technical requirements.

Kind regards,

Mei Li

Academic Editor

PLOS ONE

Additional Editor Comments (optional):

Reviewers' comments:

Reviewer's Responses to Questions

**Comments to the Author**

1. If the authors have adequately addressed your comments raised in a previous round of review and you feel that this manuscript is now acceptable for publication, you may indicate that here to bypass the “Comments to the Author” section, enter your conflict of interest statement in the “Confidential to Editor” section, and submit your "Accept" recommendation.

Reviewer #2: All comments have been addressed

2. Is the manuscript technically sound, and do the data support the conclusions?

Reviewer #2: Yes

3. Has the statistical analysis been performed appropriately and rigorously? 

Reviewer #2: Yes

4. Have the authors made all data underlying the findings in their manuscript fully available?

Reviewer #2: Yes

5. Is the manuscript presented in an intelligible fashion and written in standard English?

Reviewer #2: Yes

6. Review Comments to the Author

Reviewer #2: (No Response)

7. PLOS authors have the option to publish the peer review history of their article (what does this mean? ). If published, this will include your full peer review and any attached files.

**Do you want your identity to be public for this peer review?** For information about this choice, including consent withdrawal, please see our Privacy Policy .

Reviewer #2: No

---

## [Editor Report · Acceptance letter]

PONE-D-24-45060R1

PLOS ONE

Dear Dr. Krepsky,

I'm pleased to inform you that your manuscript has been deemed suitable for publication in PLOS ONE. Congratulations! Your manuscript is now being handed over to our production team.

Kind regards,

on behalf of

Dr Mei Li

Academic Editor

PLOS ONE